# The Safety of mRNA-1273, BNT162b2 and JNJ-78436735 COVID-19 Vaccines: Safety Monitoring for Adverse Events Using Real-World Data

**DOI:** 10.3390/vaccines10020320

**Published:** 2022-02-17

**Authors:** Soonok Sa, Chae Won Lee, Sung Ryul Shim, Hyounggyoon Yoo, Jinwha Choi, Ju Hee Kim, Kiwon Lee, Myunghee Hong, Hyun Wook Han

**Affiliations:** 1Department of Biomedical Informatics, CHA University School of Medicine, CHA University, Seongnam 13488, Korea; soonoksa@chauniv.ac.kr (S.S.); leechaewon@chauniv.ac.kr (C.W.L.); ryul01@korea.ac.kr (S.R.S.); jhkimlina@chauniv.ac.kr (J.H.K.); 2Institute of Basic Medical Sciences, School of Medicine, CHA University, Seongnam 13488, Korea; 3Institute for Biomedical Informatics, School of Medicine, CHA University, Seongnam 13488, Korea; 4Healthcare Big-Data Center, Bundang CHA Hospital, Seongnam 13488, Korea; 5Department of Health and Medical Informatics, Kyungnam University, College of Health Sciences, Changwon 51767, Korea; 6Department of Clinical Pharmacology and Therapeutics, CHA Bundang Medical Center, CHA University, Seongnam 13496, Korea; hgyoo0317@chamc.co.kr; 7Department of Pediatrics, Korea University College of Medicine, Seoul 08308, Korea; jazzina@korea.ac.kr; 8Spidercore Inc., Daejeon 34134, Korea; kiwon@spidercore.io

**Keywords:** COVID-19, vaccines, severe adverse events, safety

## Abstract

Two mRNA COVID-19 vaccines (mRNA-1273, Moderna; and BNT162b2, Pfizer-BioNTech) and one viral vector vaccine (JNJ-78436735, Janssen/Johnson and Johnson) are authorized in the US to hinder COVID-19 infections. We analyzed severe and common adverse events in response to COVID-19 vaccines using real-world, Vaccine Adverse Effect Reporting System (VAERS) data. From 14 December 2020 to 30 September 2021, 481,172 (50.7 ± 17.5 years, males 27.89%, 12.35 per 100,000 people) individuals reported adverse events (AEs). The median time to severe AEs was 2 days after injection. The risk of severe AEs following the one viral vector vaccine (OR = 1.044, 95% CI = 1.005–1.086) was significantly higher than that after the two mRNA vaccines, and the risk among males (OR = 1.374, 95% CI = 1.342–1.406) was higher than among females, except for anaphylaxis. For common AEs, however, the risk to males (OR = 0.621, 95% CI = 0.612–0.63) was lower than to females. In conclusion, we provided medical insight and clinical guidance about vaccine types by characterizing AEs using real-world data. In particular, COVID-19 mRNA vaccines are safer than viral vector vaccines with regard to coagulation disorders, whereas inflammation-related AEs are lower in the viral vaccine. The risk–benefit ratio of vaccines should be carefully considered, and close monitoring and management of severe AEs is needed.

## 1. Introduction

The World Health Organization (WHO) declared the novel coronavirus (COVID-19) outbreak a global pandemic on 11 March 2020. The coronavirus disease caused by severe acute respiratory syndrome coronavirus-2 (SARS-CoV-2) has caused approximately 243.3 million illnesses and 4.9 million deaths worldwide [1].

Safe and effective vaccines against SARS-CoV-2 are needed to stop the pandemic. Two 2-dose messenger RNA (mRNA) vaccines (mRNA-1273, Moderna; and BNT162b2, Pfizer-BioNTech) and one 1-dose viral vector vaccine (JNJ-78436735, Janssen/Johnson and Johnson) against SARS-CoV-2 have been authorized in the US to prevent serious COVID-19 infections [2,3]. After the emergence of clinical trial results, mass vaccination campaigns began in December 2020. 

Although the rapid development of COVID-19 vaccines and mass vaccination have provided effective protection against COVID-19, systematic trends in unexpected serious adverse events (AEs) associated with the vaccines have not been identified.

Rare or serious outcomes associated with vaccines might not be identified in phase 3 trials because of limited sample sizes, restrictive inclusion criteria, limited follow-up durations, and trial participants who differ in meaningful ways from the population ultimately receiving the vaccines [4,5]. Safety monitoring of AEs after vaccination is critical to ensure safety, maintain trust, and inform policy [2]. However, to our knowledge, this study is the first to use real-world big data to investigate the associations between COVID-19 vaccinations and systematically classified AEs, including death. 

The Centers for Disease Control and Prevention (CDC) and the U.S. Food and Drug Administration (FDA) conduct post-licensure vaccine safety monitoring using the Vaccine Adverse Event Reporting System (VAERS), a spontaneous or passive reporting system. After vaccine approval, the CDC and FDA continue to monitor product safety for use by collecting and analyzing spontaneous reports of AEs that occur in people following vaccination [6].

This study evaluates AEs to provide accurate post-vaccination safety information by comparing the two mRNA vaccines and one viral vector vaccine. 

## 2. Materials and Methods

### 2.1. Data

This study was performed according to the Strengthening the Reporting of Observational Studies in Epidemiology (STROBE) guidelines. We used VAERS data from 14 December 2020 to 30 September 2021 to analyze and characterize post-vaccination AEs associated with the COVID-19 vaccines authorized for the U.S. population aged 18 years and older. The VEARS was developed in 1990 as a US vaccine safety surveillance program by the Centers for Disease Control and Prevention (CDC) and the Food and Drug Administration (FDA). It collects information regarding adverse events (AEs) to serve as an early warning system for potential safety issues with US-licensed vaccines. Vaccine recipients, health care providers, and vaccine makers can openly report side effects to the VEARS [6]. VAERS data and individual reports without personally identifying information are available to the public on the VAERS (https://vaers.hhs.gov/data.html) and CDC WONDER websites (https://wonder.cdc.gov/vaers.html) (all accessed on 25 October 2021).

### 2.2. Setting and Study Population

We compared AE incidence after vaccination with either of the two mRNA vaccines (mRNA-1273, Moderna; and BNT162b2, Pfizer-BioNTech) or one viral vector vaccine (JNJ-78436735, Janssen/Johnson and Johnson), as reported in VAERS data. Our inclusion criteria were the date (14 December 2020 to 30 September 2021), age (18 years and older), and receipt of one of the three vaccines. After removing duplicates, we extracted 481,172 reports of individuals who experienced at least one AE after COVID-19 vaccination. We processed and analyzed the data using Python version 3.7.6 (Python Software Foundation, Wilmington, DE, USA) and R version 4.0.5 (R Foundation for Statistical Computing, Vienna, Austria).

### 2.3. Chosen Severe AEs

We selected 25 severe AEs, one of which was death, on the advice of a focus group of three clinical experts. We also used two prior studies to select the severe AEs considered in this study [2,7], which are listed in Appendix A.

### 2.4. Statistical Analysis

Variables with *p* < 0.05 in the univariate logistic regression analysis, along with age, sex, onset days, and vaccine type, were included in the multivariate logistic regression analysis. When multiple AEs were concurrently reported, a sensitivity analysis was performed by subdividing the number of AE reports into five levels as 1–5, 6–10, 11–15, 16–20, and >20. The data obtained were subjected to normality testing, and a two-sided *p* < 0.05 was considered statistically significant. Statistical analysis was performed using R version 4.0.5 (R Foundation for Statistical Computing, Vienna, Austria).

## 3. Results

### 3.1. Population Characteristics

The population characteristics recorded in the VAERS data are presented in Appendix A. From 14 December 2020 to 30 September 2021, 481,172 individuals (mean age 50.7 ± 17.5 years, male: 27.89%, female: 72.11%, 12.35 per 100,000 US people) were reported to have experienced AEs after COVID vaccination. The recipients of mRNA-1273 (Moderna, Cambridge, MA, USA), BNT162b2 (Pfizer-BioNTech, NY, USA), and JNJ-78436735 (Janssen/Johnson and Johnson, Beerse, Belgium) numbered 237,158 (15.66 per 100,000 people), 205,436 (9.21 per 100,000 people), and 38,578 (25.72 per 100,000 people), respectively. The data were processed using the workflow process depicted in Figure 1. The data from VAERS reflect an adult population (18 years and over) vaccinated with mRNA-1273, BNT162b2, or JNJ-78436735 during the study period.

### 3.2. Common AEs

The demographic characteristics of individuals who reported common AEs are presented in Table 1. The total reported incidence of common AEs was 902,889 (23 per 100,000 people). The incidence following vaccination with mRNA-1273, BNT162b2, and JNJ-78436735 was 474,342 (31.31 per 100,000 people), 342,913 (15.17 per 100,000 people), and 85,634 (57.09 per 100,000 people), respectively. The common AEs consisted of 9766 symptoms, excluding 25 severe AEs (138 symptoms). The top-25 most common AEs among the recipients of one of the two mRNA vaccines or one viral vector vaccine are presented in Appendix A. The most common AEs following administration of each vaccine were headache, fatigue, pyrexia, chills, pain, dizziness, and nausea. Among those seven common AEs, the incidence after the one viral vector vaccine was about twice that of the two mRNA vaccines, and the risk of common AEs with the two mRNA vaccines was higher after the second dose than after the first (Appendix A).

The incidence of COVID-19 breakthrough infection was about three times higher after receiving the one viral vector vaccine than after receiving one of the two mRNA vaccines. Among the COVID-19 breakthrough infections in our data, the odds ratio (OR) of males to females was 1.527 (CI = 1.463–1.593), and the odds ratio by vaccine type was significantly higher for the one viral vector vaccine (OR = 1.396, CI = 1.307–1.490, *p* < 0.001) than for either of the two mRNA vaccines. In subgroup analysis of internal comparisons for dysmenorrhea or disruption of menstrual cycle symptom AEs, the OR of mRNA-1273 to BNT162b2 was 0.593 (CI = 0.561–0.628), and the OR of JNJ-78436735 to BNT162b2 was 0.992 (CI = 0.909–1.083) (Appendix A). Between the two mRNA vaccines, the odds ratio of mRNA-1273 to BNT162b2 for a COVID-19 breakthrough infection was 0.535 (CI = 0.509–0.562, data not shown). Significant associations were detected between AE incidence and sex, age, onset day, and vaccine type (Appendix A). JNJ-78436735 showed an AE incidence two to eight times higher than that of the other vaccines in pruritus, injection site erythema, injection site swelling, erythema, and injection site pruritus. The incidence of hyperhidrosis after mRNA-1273 was more than three times higher than that after the other vaccines. The mRNA vaccines carried relatively high risks for local pain, and the viral vector vaccine carried relatively high risks for systemic pain (Appendix A).

### 3.3. Severe AEs

The demographic characteristics of individuals who reported severe AEs following vaccination are presented in Table 2. The total reported incidence of severe AEs was 35,619 cases (0.9 per 100,000 people). The incidence after vaccination with mRNA-1273, BNT162b2, and JNJ-78436735 was 15,438 (1.0 per 100,000 people), 16,918 (0.75 per 100,000 people), and 3263 (2.18 per 100,000 people), respectively. The incidences of severe AEs per 100,000 people among recipients of one of the two mRNA vaccines or one viral vector vaccine are illustrated in Appendix A.

The incidence (per 100,000 people) of the 25 severe AEs and the onset days after vaccination are compared in Table 2. The severe AEs with the largest differences between the two mRNA vaccines and one viral vector vaccine were cerebral venous sinus thrombosis (CVST) and Guillain–Barré syndrome (GBS), and those incidences were approximately 10-fold higher after the viral vector vaccine than the mRNA vaccines (Table 2). The most frequently reported severe AE after all the mRNA vaccines was lymphadenopathy. Most of the severe AEs varied significantly by sex, age, onset day, and vaccine type. When we calculated odds ratios, the risk of most severe AEs was higher in males than in females and higher after the one viral vector vaccine than after either of the two mRNA vaccines.

After mRNA-1273, BNT162b2, and JNJ-78436735, respective numbers of 79, 79, and 13 cases of anaphylaxis were reported. Although JNJ-78436735 had the fewest number of reports, it also had the lowest number of administered cases; its incidence value was 2–3 times higher than that of the mRNA vaccines. Anaphylaxis was acute on the day of vaccination, with an onset day (median) of 0 (Table 2). In our univariate analysis for anaphylaxis, the odds ratio of the one viral vector vaccine to the two mRNA vaccines was 1.18 (data not shown). However, in our multivariate regression analysis adjusted for sex, age, and onset of symptoms as covariates, the odds ratio of the one viral vector vaccine to the two mRNA vaccines was 0.619 (CI = 0.495–0.772) (Appendix A), confirming that the mRNA vaccines carried a higher risk for anaphylaxis. In addition, we found an association with sex in multivariate regression analysis: the odds ratio was significantly lower in males than in females, which was in contrast to our finding for severe AEs overall (Appendix A).

Following mRNA-1273, BNT162b2, and JNJ-78436735 administration, death was reported in 2286, 2005, and 447 cases, respectively. Although JNJ-78436735 reported fewer cases, the incidence per 100,000 people was 2–3 times higher than that after the mRNA vaccines. The incidence of death declined in the order of JNJ-78436735, mRNA-1273, and BNT162b2 (Table 2). In our analysis, we detected associations by sex and vaccine type. The odds ratio of males to females was 2.761 (CI = 2.599–2.933), and the odds ratio of the viral vector vaccine to the mRNA vaccines was 1.901 (CI = 1.713–2.111) (Appendix A).

Next, we compared the incidence of 25 severe AEs between young (18–64 years) and older people (≥65 years). Among 25 severe AEs, 19 were significantly associated with age class (young vs. older people; *p* < 0.05) (Table 3). The incidence of 13 severe AEs of hemorrhagic stroke, ischemic stroke, pulmonary embolism, acute respiratory distress syndrome, acute myocardial infarction, anemia, lymphopenia, neutropenia, other thrombosis, thrombocytopenia, deep vein thrombosis, multisystem inflammatory syndrome, and death was significantly high (*p* < 0.05) in older people compared to young people. Older people (≥65 years) had a 7.84 times higher incidence of death by post-vaccination AEs than younger people did (Table 3).

#### 3.3.1. Central Nervous Disorders

Severe AEs of the central nervous system differed after the two mRNA vaccines and one viral vector vaccine. Our analysis focused on six major, severe AEs. The AEs with the greatest difference among vaccines were CVST and GBS, and their incidence after the one viral vector vaccine was about 10–15 times higher than that after either of the two mRNA vaccines.

Cases of convulsion/seizures were observed mainly on the day of vaccination. For CVST, the median onset occurred at 10.5 to 16.5 days after injection.

The incidence of Bell’s palsy after the mRNA vaccines was about half that seen after the viral vector vaccine. In addition, the onset of Bell’s palsy after either of the two mRNA vaccines occurred at seven to eight days after injection, much sooner than the 16 days seen with the viral vector vaccine. According to the results of our multivariate regression analysis for Bell’s palsy, the odds ratio for one of the mRNA vaccines was significantly lower than that for the viral vector vaccine, but the difference was not statistically significant (Appendix A). However, an association with sex was detected. The odds ratio of males to females for Bell’s palsy was 2.042 (CI = 1.885–2.213), indicating a higher risk in males than in females (Appendix A). On the other hand, between the two mRNA vaccines, the OR of mRNA-1273 (OR = 0.714, CI = 0.578–0.882, *p* = 0.002) was significantly low compared to that of the BNT162b2 (Appendix A)

According to our analysis, the incidence of GBS was about 10 times higher after the viral vector vaccine than after one of the mRNA vaccines. With all vaccines, onset (median) of GBS occurred after 8 to 14 days (Table 2). We detected causal associations with sex and vaccine type in GBS. Males had twice the odds ratio of females (OR = 2.310, CI = 1.931–2.764), and the viral vector vaccine had a remarkably high odds ratio of 4.183 (CI = 3.411–5.130) compared with the two mRNA vaccines (Appendix A).

The incidence of CVST was associated with vaccine type and onset of symptoms (number of days). The odds ratio of the viral vector vaccine to mRNA vaccines was significantly high (OR = 5.049, CI = 3.492–7.301, *p* < 0.001), and the odds ratio of symptom onset (days) was significantly high (OR = 1.0125, CI = 1.009–1.016, *p* < 0.001) (Appendix A). There was no significant difference between any two mRNA vaccines (*p* = 0.668; Appendix A).

The incidence of both hemorrhagic and ischemic stroke was 4–6 times higher after the viral vector vaccine than after either mRNA vaccine.

The severe AE with the highest incidence was convulsions/seizures. The incidence after the viral vector vaccine was 4–5 times higher than after the mRNA vaccines. Convulsions/seizures were observed on the same day as vaccination (Table 2).

The incidence of acute disseminated encephalomyelitis was 3 times higher after the viral vector vaccine than after either of the two mRNA vaccines. For mRNA-1273 and JNJ-78436735, onset (median) after injection was 12.5 days (Table 2). However, after BNT162b2, the onset (median) for acute disseminated encephalomyelitis occurred at 3.5 days.

#### 3.3.2. Respiratory Disorders

The incidence of pulmonary embolism and acute respiratory distress syndrome was about 5–7 times higher after the one viral vector vaccine than after either of the mRNA vaccines. After the mRNA vaccines, the incidence was higher after the second dose than after the first (Appendix A). The onset (median) of acute respiratory distress syndrome was 42 days after administration, but it was widely observed from two to six weeks after vaccination (Table 2). The odds ratio of acute respiratory distress syndrome for males was twice that for females, and the odds ratio of the viral vector vaccine to the mRNA vaccines was 2.951 (CI = 1.706–5.105) (Appendix A). There was no significant difference between mRNA-1273 and BNT162b2 (*p* = 0.581; Appendix A).

#### 3.3.3. Cardiac Disorders

The incidence of myocarditis/pericarditis after the two mRNA vaccines was similar to that after the one viral vector vaccine. With the mRNA vaccines, the incidence of cardiac AEs was higher after the second dose than the first dose (Table 2, Appendix A). The odds ratio of males to females was significantly high (OR = 5.614, CI = 5.074–6.211, *p* < 0.001) (Appendix A). The OR of the viral vector vaccine to the mRNA vaccines was significantly low (OR = 0.378, CI = 0.305–0.469, *p* < 0.001) (Appendix A). Additionally, the OR of mRNA-1273 was significantly low (OR = 0.816, CI = 0.739–0.901, *p* < 0.001) compared to BNT162b2 (Appendix A).

#### 3.3.4. Gastrointestinal Disorders

The incidence of appendicitis was about 2 times higher after the viral vector vaccine than after either mRNA vaccine. The median number of days to onset for all vaccines was 3 to 5.4 (Table 2).

#### 3.3.5. Hematologic Disorders

The incidence of deep vein thrombosis was 7–9 times higher after the one viral vector vaccine than after either mRNA vaccine. The incidence of thrombocytopenia was about 4–6 times higher after the one viral vector vaccine than the two mRNA vaccines, and the incidence other thrombosis events was about 6–8 times higher after the viral vector vaccine than the mRNA vaccines (Table 2). Of the three hematologic disorders, deep vein thrombosis showed significantly high OR of males to females (OR = 2.103, CI = 1.910–2.315, *p* < 0.001). The OR of the viral vector vaccine to the mRNA vaccines was significantly high (OR = 4.015, CI = 3.588–4.493) (Appendix A), while the OR of mRNA-1273 was significantly low (OR = 0.772, CI = 0.690–0.863, *p* < 0.001) compared to BNT162b2 (Appendix A). In addition, the most frequent severe AEs following administration of all vaccines was lymphadenopathy, and its OR was significantly low (OR = 0.314, CI = 0.284–0.346, *p* < 0.001) in the viral vector vaccine compared to the mRNA vaccines (Appendix A). Between two mRNA vaccines, the odds ratio of mRNA-1273 was significantly low (OR = 0.792, CI = 0.765–0.819, *p* < 0.001) compared to BNT162b2 (Appendix A). We also detected the significant association of lymphadenopathy by sex. The odds ratio for males to females was 0.524 (CI = 0.501–0.548).

#### 3.3.6. Correlation Analysis between Death and Severe AEs

In a correlation analysis between post-vaccination death and the remaining 24 severe AEs, acute respiratory distress syndrome, hemorrhagic stroke, and acute myocardial infarction were the most significantly correlated with post-vaccination mortality (r = 0.053, r = 0.032, and r = 0.028, respectively; all *p* < 0.001). In addition, multivariate logistic regression analysis for death after adjusting for sex (reference: female), age, symptom onset (number of days), and vaccine type (reference: BNT162b2) with severe AEs as covariates revealed that acute respiratory distress syndrome (OR = 20.510, CI = 12.620–33.332, *p* < 0.001), hemorrhagic stroke (OR = 9.965, CI = 6.406–15.499, *p* < 0.001), and acute myocardial infarction (OR = 3.872, CI = 2.794–5.366, *p* < 0.001) were most significantly associated with post-vaccination mortality (Table 4).

### 3.4. Serious Outcomes

The characteristics of the serious outcomes following the three vaccines are presented in Appendix A. Serious outcomes, both disability and death, from the one viral vector vaccine (JNJ-78436735) were 2–3.5 times more frequent than those from the mRNA vaccines. For each vaccine, the outcomes were divided into five levels according to concurrent AE number (Table 5). The proportions of the five levels were 1–5 (73.08%), 6–10 (21.32%), 11–15 (4.03%), 16–20 (0.96%), and >20 (0.61%). The largest proportion at 73.08% was observed for the 1–5 outcomes and the smallest proportion was the 0.61% of the >21 outcomes.

## 4. Discussion

In this study, we are the first to use VAERS data to systemically analyze severe and common AEs following administration of COVID-19 vaccines. Overall, the risks of the mRNA vaccines were significantly lower than those of the one viral vector vaccine except for anaphylaxis. Although males had a higher risk of severe AEs than females, they had a lower risk of common AEs. Our findings support caution about severe AEs and mortality in the post-vaccination period. The risk of severe AEs was particularly high after the viral vector vaccine with regard to death, Guillain–Barré syndrome, acute respiratory distress syndrome, and coagulation disorders (cerebral venous sinus thrombosis, pulmonary embolism, thrombocytopenia, deep vein thrombosis), which might have implications for monitoring strategies.

Most common AEs were associated with sex, age, onset day, and vaccine type. Our odds ratio analyses confirmed that the risk of most common AEs was significantly higher in females than in males and higher after the one viral vector vaccine than after either of the two mRNA vaccines. The trends in comparative incidences of each common AE per 100,000 people following the three types of vaccines were similar to those reported in a previous study [8]. In this study, the risk of a breakthrough COVID-19 infection was higher in males than in females and higher after receiving the one viral vector vaccine than either of the two mRNA vaccines. For breakthrough COVID-19 infections following the two mRNA vaccines, the odds ratio of females to males was low, and the odds ratio of mRNA-1273 compared to BNT162b2 was 0.5694, indicating that mRNA-1273 had a lower risk than BNT162b2. This result is the same as previous studies reporting that mRNA-1273 vaccine was more effective, and mRNA-1273 had a lower incidence rate of COVID-19 breakthrough infections than BNT162b2 [9]. In addition, according to the Asorsi et al.’s study [10] compared to the case without vaccination, the mRNA-1273 (OR = 0.045, CI = 0.038–0.053; OR = 0.280, CI = 0.260–0.310) was lower than BNT162b2 (OR = 0.077, CI = 0.070–0.086; OR = 0.350, CI = 0.320–0.380) for the Delta variant and Omicron variant, respectively. This might have implications for guidance to high-risk groups about the different vaccine brands.

We found that most severe AEs were associated with sex, age, onset day, and vaccine type. Our odds ratio analyses confirmed that males were at a significantly higher risk of most severe AEs than females, and the viral vector vaccine carried significantly more risk than the mRNA vaccines. Because the onset days (median) of severe AEs varied by vaccine type, careful monitoring after vaccination needs to be differentiated by vaccine type.

This study demonstrated that AEs in thrombosis, heart, hematology, and nervous system were high in older people, and supported previous studies [11]. These results suggested that the elderly may need close monitoring of AEs after vaccination. On the other hand, some AEs, including Bell’s palsy, myocarditis/pericarditis, convulsions/seizures, lymphadenopathy, and anaphylaxis, exhibited a high incidence in young people. In the previous studies, lymphadenopathy [12] and myocarditis/pericarditis [13] have been reported with a higher frequency in the younger population. This study also showed more frequency of inflammation-related AEs such as Bell’s palsy, myocarditis/pericarditis, lymphadenopathy, which seemed to estimate high in young people.

In this study, we found that the ORs of mRNA-1273 for severe AEs such as GBS, MT, DVT, and LP were significantly low compared to BNT162b2. The present study similarly found that the serious AEs of mRNA-1273 vaccination were slightly lower than BNT162b2 (49.0% versus 49.8%) [14]. On the other hand, Beatty A.L. et al. [15] showed that the mRNA-1273 vaccine increased ORs 1.88 times (95% CI, 1.63–2.17) the odds of severe or very severe AEs (*p* < 0.001) compared to BNT162b2 vaccine.

The mortality rate for the COVID-19 vaccine was 0.002%, slightly higher than that of the traditional flu vaccine in 2019, which was 0.0018% [16]. Although the mortality due to AEs after vaccination is very low, mortality was highly correlated with acute respiratory distress syndrome, hemorrhagic stroke, and acute myocardial infarction, which may suggest the need for close monitoring. Because days to death varied by vaccine type (mRNA vaccines after 5 days and viral vector vaccine after 10 days), monitoring should be differentiated according to vaccine type. The risk of death from the viral vector vaccine was higher than that from the mRNA vaccine, and males were at higher risk of vaccine-related death than females, which might have implications for monitoring strategies, especially in high-risk groups.

Previous studies reported that vaccine-associated anaphylaxis is slightly more common in females than males [17,18], and that the sexual disproportion is quite notable with the mRNA COVID-19 vaccines [17,18]. Although anaphylaxis after COVID-19 vaccination is rare, vaccine providers need to offer careful and close monitoring because anaphylaxis is a potentially life-threatening medical emergency after mRNA COVID-19 vaccination.

GBS has been reported as an AE following conventional vaccines, including annual influenza vaccines [19]. Cases of GBS after vaccination for COVID-19 have also been reported [20]. The fact sheet [21] for the mRNA-1273 vaccine mentions the possibility of developing GBS. Autoantibody-mediated immunological damage of peripheral nerves via a mechanism of molecular mimicry between structural components of peripheral nerves and the microorganism has been hypothesized as a possible explanation for vaccine-associated GBS [22]. The risk of GBS was remarkably higher in males than females after both the mRNA vaccines and the viral vector vaccine [23]. Our findings support those of previous studies and imply that close monitoring for GBS is necessary, especially in males.

CVST has been reported as an AE after COVID-19 vaccination [24,25]. A previous study reported that the risk of CVST after viral vector vaccination was higher than after mRNA vaccination, and that the risk was especially high for females [26]. Consistently, we found that the incidence rate and odds ratio were higher after the viral vector vaccine than the mRNA vaccines and in females than males. These findings support previous studies and imply that close monitoring for CVST is necessary, especially after viral vector COVID-19 vaccination in females.

Recently, myocarditis and pericarditis have been reported as rare AEs of COVID-19 mRNA vaccinations, especially in young adult and adolescent males [27]. Although we analyzed only adults (older than 18), our results are similar to those of the previous study. The median age of the population group that experienced myocarditis or pericarditis was relatively young at 35 years. The risk of myocarditis or pericarditis was very high for young men, and the risk with the mRNA vaccines was much higher than that with the viral vector vaccine, which imply the need for close monitoring for myocarditis and pericarditis, especially in young males, after mRNA COVID-19 vaccination.

### Limitations

Although our study results, which we compiled from a large collection of real-world data, show patterns similar to those in previous reports, generalizability might be limited due to the characteristics of self-reported data. Most VAERS data are voluntarily self-reported, which means they could be subject to bias. Therefore, more attention should be paid to explaining a direct causal relationship. Our results require further validation by independent studies. In addition, this study has limitations in interpretation of the results because personal health information, such as height, weight, underlying diseases, and medications, cannot be taken into account. Only the age and sex of the target population are available. Appendix A show the results of subgroup analysis of internal comparisons (within those who have reported AEs to VAERS) using ORs, and thus the reported AEs of each vaccine should be interpreted with caution. The current analysis for COVID-19 vaccines did not address long-term AEs. The SARS-CoV2 variant was not considered in this study. Time series analysis considering mutation prevalence is needed in the future. Future studies should address the relationships between COVID-19 vaccines and individual biological characteristics, especially how underlying diseases and medications affect the AEs that follow vaccination.

## 5. Conclusions

In conclusion, we provided medical insight and clinical guidance about vaccine types by characterizing AEs using real-world data. In particular, COVID-19 mRNA vaccines are safer than viral vector vaccines with regard to coagulation disorders, whereas inflammation-related AEs are lower in the viral vaccine. The risk–benefit ratio of vaccines should be carefully considered, and close monitoring and management of severe AEs is needed.

## Figures and Tables

**Figure 1 vaccines-10-00320-f001:**
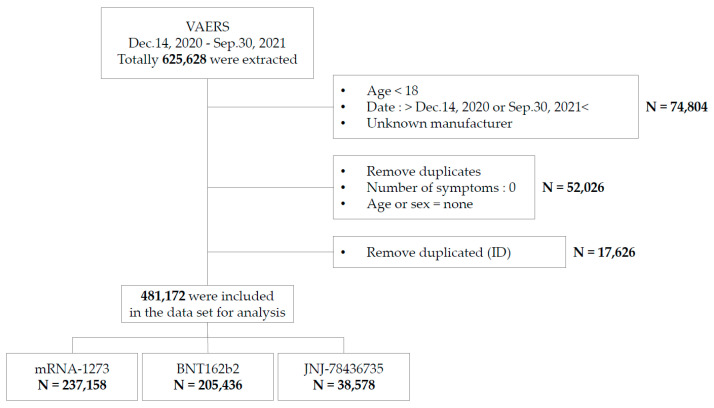
Workflow of this study.

**Table 1 vaccines-10-00320-t001:** Demographic characteristics of individual cases with specific AEs among recipients of one of the two mRNA vaccines (BNT162b2, Pfizer-BioNTech; and mRNA-1273, Moderna) or one viral vector vaccine (JNJ-78436735, Janssen/Johnson and Johnson).

	Severe AEs				Common AEs			
	mRNA-1273	BNT162b2	JNJ-78436735	Sum	mRNA-1273	BNT162b2	JNJ-78436735	Sum
Female								
18–24	454	565	118	1137	16,802	15,772	5908	38,482
25–39	2492	2983	378	5853	88,919	75,605	19,045	183,569
40–49	1914	2445	316	4675	69,034	56,110	12,668	137,812
50–64	2596	2762	509	5867	106,474	75,185	17,454	199,113
65–74	1388	1231	216	2835	65,980	29,710	3602	99,292
≥75	1196	1017	208	2421	28,884	12,201	980	42,065
Sum (sex ratio, %)	10,040 (51.8)	11,003 (49.5)	1745 (51.7)	22,788	376,093 (51.2)	264,583 (47.6)	59,657 (43.8)	700,333
Male								
18–24	418	591	128	1137	5409	5929	3957	15,295
25–39	966	1244	316	2526	21,342	21,565	8934	51,841
40–49	657	827	255	1739	14,487	14,012	4417	32,916
50–64	1249	1385	497	3131	26,787	19,985	6580	53,352
65–74	1033	928	191	2152	19,358	10,922	1541	31,821
≥75	1075	940	131	2146	10,866	5917	548	17,331
Sum (sex ratio, %)	5398 (55.8)	5915 (52.5)	1518 (50.6)	12,831	98,249 (52.9)	78,330 (48.7)	25,977 (41.8)	202,556
Total * (per 100,000 people)	15,438 (10.191)	16,918 (7.485)	3263 (21.753)	35,619 (9.075)	474,342 (313.135)	342,913 (151.709)	85,634 (570.881)	902,889 ^†^ (230.026)

* From 14 December 2020 to 30 September 2021, mRNA-1273 (Moderna) was administered to 151,481,614 individuals, BNT162b2 (Pfizer-BioNTech) was administered to 226,033,301 individuals, and JNJ-78436735 (Janssen/Johnson and Johnson) was administered to 15,000,326 individuals according to the CDC COVID-19 tracker. ^†^ A total of 902,889 adverse events were reported from 481,172 individuals.

**Table 2 vaccines-10-00320-t002:** Incidence (per 100,000 people) of 25 severe AEs with onset days after vaccination.

	mRNA-1273		BNT162b2		JNJ-78436735	
Symptom	Number	Incidence of Events (Onset Day-Median)	Number	Incidence of Events (Onset Day-median)	Number	Incidence of Events (Onset Day-Median)
Bell’s palsy	1093	0.72 (8)	1234	0.55 (7)	204	1.36 (16)
Stroke, hemorrhagic	77	0.05 (7)	82	0.04 (7)	36	0.24 (5)
Stroke, ischemic	302	0.20 (7)	334	0.15 (5)	100	0.67 (10)
Encephalitis/myelitis/encephalomyelitis	30	0.02 (6)	33	0.01 (10.5)	6	0.04 (9)
Cerebral venous sinus thrombosis	44	0.03 (16.5)	47	0.02 (16.5)	46	0.31 (10.5)
Convulsions/seizures	1479	0.98 (0)	1664	0.74 (0)	574	3.83 (0)
Guillain–Barré syndrome	13	0.01 (8)	15	0.01 (14.5)	145	0.97 (14)
Transverse myelitis	53	0.03 (12)	57	0.03 (8)	20	0.13 (13)
Acute disseminated encephalomyelitis	6	0.00 (12.5)	13	0.01 (3.5)	4	0.03 (11.5)
Narcolepsy/cataplexy	12	0.01 (1)	14	0.01 (1)	2	0.01 (0)
Pulmonary embolism	857	0.57 (9)	843	0.37 (10)	428	2.85 (13)
Acute respiratory distress syndrome	42	0.03 (12.5)	45	0.02 (43)	17	0.11 (24.5)
Acute myocardial infarction	230	0.15 (5)	287	0.13 (8)	43	0.29 (10)
Myocarditis/pericarditis	791	0.52 (3)	1 072	0.47 (3)	93	0.62 (8)
Appendicitis	160	0.11 (3)	291	0.13 (5.5)	38	0.25 (4)
Anemia	37	0.02 (11)	39	0.02 (6)	7	0.05 (16)
Lymphadenopathy	6761	4.46 (1)	7847	3.47 (1)	420	2.80 (1)
Lymphopenia	13	0.01 (16.5)	17	0.01 (4)	3	0.02 (8)
Neutropenia	18	0.01 (3)	31	0.01 (10.5)	6	0.04 (1.5)
Other thrombosis	354	0.23 (11)	358	0.16 (9)	202	1.35 (13)
Thrombocytopenia	261	0.17 (10.5)	302	0.13 (10.5)	114	0.76 (13)
Deep vein thrombosis	643	0.42 (10)	716	0.32 (8)	448	2.99 (12)
Anaphylaxis	79	0.05 (0)	79	0.03 (0)	13	0.09 (0)
Multisystem inflammatory syndrome in children/adults	1	0.00 (6)	2	0.00 (4)	1	0.01 (0)
Death	2286	1.51 (5)	2005	0.89 (5)	447	2.98 (10)

**Table 3 vaccines-10-00320-t003:** Incidence (per 100,000 people) of 25 severe AEs between young (18–64 years) and older people (≥65 years) after vaccination.

	Young People (18–64 Years)		Older People (≥65 Years)		Total	
Symptom	Number	Incidence ^†^ of Events	Number	Incidence ^†^ of Events	Number	Incidence ^†^ of Events
Bell’s palsy **	1889	0.707	616	0.622	2505	0.684
Stroke, hemorrhagic ***	96	0.036	84	0.085	180	0.049
Stroke, ischemic ***	338	0.126	364	0.368	702	0.192
Encephalitis/myelitis/encephalomyelitis	49	0.018	25	0.025	74	0.020
Cerebral venous sinus thrombosis	106	0.040	27	0.027	133	0.036
Convulsions/seizures ***	3287	1.229	574	0.58	3861	1.054
Guillain–Barré syndrome	355	0.133	155	0.157	510	0.139
Transverse myelitis	90	0.034	35	0.035	125	0.034
Acute disseminated encephalomyelitis	17	0.006	5	0.005	22	0.006
Narcolepsy/cataplexy	18	0.007	7	0.007	25	0.007
Pulmonary embolism ***	1242	0.465	804	0.812	2046	0.559
Acute respiratory distress syndrome ***	36	0.013	56	0.057	92	0.025
Acute myocardial infarction ***	240	0.090	254	0.257	494	0.135
Myocarditis/pericarditis ***	1573	0.588	193	0.195	1766	0.482
Appendicitis ***	389	0.145	64	0.065	453	0.124
Anemia ***	43	0.016	34	0.034	77	0.021
Lymphadenopathy ***	13,337	4.988	1681	1.699	15,018	4.100
Lymphopenia *	14	0.005	13	0.013	27	0.007
Neutropenia **	25	0.009	22	0.022	47	0.013
Other thrombosis ***	519	0.194	335	0.339	854	0.233
Thrombocytopenia ***	347	0.130	252	0.255	599	0.164
Deep vein thrombosis ***	1141	0.427	592	0.598	1733	0.473
Anaphylaxis ***	1401	0.524	209	0.211	1610	0.439
Multisystem inflammatory syndrome in children/adults ***	62	0.023	48	0.049	110	0.030
Death ***	1176	0.440	3413	3.449	4589	1.253

All associations were calculated using chi-squared test. From 14 December 2020 to 30 September 2021, vaccinations for 18-year-olds and older were administered to 366,325,674 individuals, and for 65-year-olds and older they were administered to 98,962,216 individuals, according to the CDC COVID-19 tracker. * *p*-value < 0.05, ** *p*-value < 0.01, *** *p*-value < 0.001. ^†^ Incidence per 100,000 people.

**Table 4 vaccines-10-00320-t004:** Correlation analysis and multivariate logistic regression analysis between death and severe AEs.

Severe AEs	Correlation with Death (r *)	*p* Value *	OR (CI) ^†^	*p* Value ^†^
Bell’s palsy	−0.007	<0.001	-	0.859
Hemorrhagic stroke	0.032	<0.001	9.965 (6.406–15.499)	<0.001
Ischemic stroke	0.005	<0.001	0.899 (0.541–1.496)	0.683
Encephalitis/myelitis/encephalomyelitis	0.002	0.122	1.565 (0.365–6.717)	0.547
Cerebral venous sinus thrombosis	0.001	0.514	1.537 (0.368–6.423)	0.556
Convulsions seizures	0.007	<0.001	2.437 (1.877–3.164)	<0.001
Guillain Barre syndrome	0.001	0.605	0.743 (0.328–1.683)	0.476
Transverse myelitis	−0.002	0.273	-	0.887
Acute disseminated encephalomyelitis	−0.001	0.645	-	0.903
Narcolepsy cataplexy	−0.001	0.624	-	0.887
Pulmonary embolism	0.023	<0.001	2.363 (1.885–2.961)	<0.001
Acute respiratory distress syndrome	0.053	<0.001	20.510 (12.620–33.332)	<0.001
Acute myocardial infarction	0.028	<0.001	3.872 (2.794–5.366)	<0.001
Myocarditis/pericarditis	−0.001	0.651	1.453 (0.863–2.448)	0.160
Appendicitis	−0.002	0.108	0.333 (0.046–2.406)	0.276
Anemia	0.006	<0.001	3.797 (1.330–10.839)	0.013
Lymphadenopathy	−0.017	<0.001	0.149 (0.078–0.287)	<0.001
Neutropenia	0.006	<0.001	3.622 (1.069–12.273)	0.039
Other thrombosis	0.011	<0.001	1.910 (1.297–2.812)	0.001
Thrombocytopenia	0.014	<0.001	2.141 (1.427–3.213)	<0.001
Deep vein thrombosis	0.003	0.064	0.641 (0.417–0.985)	0.042
Anaphylaxis	−0.005	0.002	0.388 (0.124–1.211)	0.103
Multisystem inflammatory syndrome	0.006	<0.001	2.053 (0.775–5.439)	0.148

* *p*-value was calculated by Pearson’s correlation test. ^†^ The odds ratio and *p*-value were calculated by multivariate logistic regression analysis for death after adjusting for sex (reference: female), age, symptom onset (number of days), and vaccine type (reference: BNT162b2), each severe AE as covariates.

**Table 5 vaccines-10-00320-t005:** Number of outcomes for individuals according to 5 levels of number of adverse events for each vaccine.

Number of Outcomes	mRNA-1273(%)	BNT162b2 (%)	JNJ-78436735(%)	Sum
1~5	174,347 (73.52%)	151,908 (73.94%)	25,407 (65.86%)	351,662 (73.08%)
6~10	51,026 (21.52%)	41,618 (20.26%)	9924 (25.72%)	102,568 (21.32%)
11~15	8700 (3.67%)	8380 (4.08%)	2293 (5.94%)	19,373 (4.03%)
16~20	1921 (0.81%)	2124 (1.03%)	581 (1.51%)	4626 (0.96%)
≥21	1164 (0.49%)	1406 (0.68%)	373 (0.97%)	2943 (0.61%)
Sum	237,158	205,436	38,578	481,172

Sensitivity analysis was performed by subdividing the number of AE reports into five levels as 1–5, 6–10, 11–15, 16–20, and >20.

## Data Availability

All data analyzed in this study are public data from the Vaccine Adverse Event Reporting System (VAERS). VAERS is co-administered by the Centers for Disease Control and Prevention (CDC) and the U.S. Food and Drug Administration (FDA): https://vaers.hhs.gov/data/datasets.html (accessed on 25 October 2021).

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
