# Peer review of "The Safety of mRNA-1273, BNT162b2 and JNJ-78436735 COVID-19 Vaccines: Safety Monitoring for Adverse Events Using Real-World Data"

_vaccines, 2022, doi:10.3390/vaccines10020320_

Round 1

Reviewer 1 Report

The authors analyzed severe and common adverse events for the 2 mRNA COVID-19 vaccines Moderna and Pfizer-BioN Tech and for one viral vector vaccine Janssen/Johnson and Johnson using real-world, Vaccine Adverse Effect Reporting System (VAERS) data, based on data collected between December 14, 2020, and September 30, 2021. A total of 481,172 - 12.35 per 100,000 people - individuals reported adverse events. And the authors conclude that the risk of severe AEs following the one viral vector vaccine (OR=1.044) was significantly higher than that after the two mRNA vaccines. Consequently, the two mRNA vaccines would be safer than the one viral vector vaccine. The authors conclude that these findings must have implications for monitoring strategies. The work is interesting but needs major modifications especially in the presentation of the results. Please see attached the document with all my comments.

Reviewer 2 Report

The article with title "The Safety of COVID-19 mRNA Vaccines Compared with a Viral Vector Vaccine: Safety Monitoring for Adverse Events Using Real-World Data", authors by Soonok Sa and Coleagues provides a comprehensive list of Adverse Events following vaccination with mRNA or viral vector vaccines. Given the importance of containment of COVID-19 pandemic, there is immediate necessity for studies like the one presented by the authors.   The data are convincing and well analyzed. I have the following comments that will    1) The authors should carefully describe the difference between the presented data in table S1 and Table 1. In table S1 a population of 481172 people is presented. In table 1 a total of 902889 cases are described. The authors should make clear in the whether these cases correspond to the 481172 individuals.   2)  I believe that Table 4 should be moved into supplementary data, since it is interpreted in the text and is presents only statistically significance data.   3) The text is well written and I have detected only few grammatical and structural errors. The authors should consider revising for an optimal presentation of their manuscript. 

Round 2

Reviewer 1 Report

The authors have taken into account my comments and corrected the manuscript as required. However, the results added especially concerning the AE according to the mRNA vaccine, by making difference between the Pfizer and the Moderna vaccine have not been taken into account in the conclusion. In the light of the results added, the initial conclusion of the paper is wrong and has necessarily to be corrected. Overall severe AE are not not more frequent with JNJ than with mRNA vaccine and it depends on the type of mRNA vaccine, and the type of AE. Clearly, authors do not have to gather even in the conclusion AE caused by the two mRNA vaccines. It is clear in the table S6 that the OR for Overall severe AE is higher with Pfizer vaccine. I have also other comments that I included in the joined document, my comments are typed in red. 

Round 3

Reviewer 1 Report

The authors have taken in consideration all my comments and have corrected the manuscript as required. I think it is now clearer for the reader. I suggest some final minor corrections but important for the reader, before recommending the paper for publication.
